# Trichothecin Inhibits Cancer-Related Features in Colorectal Cancer Development by Targeting STAT3

**DOI:** 10.3390/molecules25102306

**Published:** 2020-05-14

**Authors:** Xin Qi, Meng Li, Xiao-min Zhang, Xiu-fen Dai, Jian Cui, De-hai Li, Qian-qun Gu, Zhi-hua Lv, Jing Li

**Affiliations:** 1Key Laboratory of Marine Drugs, Chinese Ministry of Education, School of Medicine and Pharmacy, Ocean University of China, Qingdao 266003, China; qxhin@163.com (X.Q.); yxzxlmlf@163.com (M.L.); xiaominzhang91@163.com (X.-m.Z.); 21180811061@stu.ouc.edu.cn (X.-f.D.); 16111520012@fudan.edu.cn (J.C.); dehaili@ouc.edu.cn (D.-h.L.); guqianq@ouc.edu.cn (Q.-q.G.); 2Open Studio for Druggability Research of Marine Natural Products, Pilot National Laboratory for Marine Science and Technology (Qingdao), Qingdao 266237, China; 3Laboratory for Marine Drugs and Bioproducts of Qingdao National Laboratory for Marine Science and Technology, Qingdao 266237, China

**Keywords:** cancer, trichothecin, STAT3, inhibitor, HCT 116

## Abstract

Signal transducer and activator of transcription 3 (STAT3) is a transcription factor that contributes to cancer progression through multiple processes of cancer development, which makes it an attractive target for cancer therapy. The IL-6/STAT3 pathway is associated with an advanced stage in colorectal cancer patients. In this study, we identified trichothecin (TCN) as a novel STAT3 inhibitor. TCN was found to bind to the SH2 domain of STAT3 and inhibit STAT3 activation and dimerization, thereby blocking STAT3 nuclear translocation and transcriptional activity. TCN did not affect phosphorylation levels of STAT1. TCN significantly inhibited cell growth, arrested cell cycle at the G0/G1 phase, and induced apoptosis in HCT 116 cells. In addition, the capacities of colony formation, migration, and invasion of HCT 116 cells were impaired upon exposure to TCN with or without IL-6 stimulation. In addition, TCN treatment abolished the tube formation of HUVEC cells in vitro. Taken together, these results highlight that TCN inhibits various cancer-related features in colorectal cancer development in vitro by targeting STAT3, indicating that TCN is a promising STAT3 inhibitor that deserves further exploration in the future.

## 1. Introduction

Signal transducers and activators of transcription 3 (STAT3) is an important member of the family of signal transducers and activators of transcription, which participates in relaying signals from cytokines and growth factors [1]. STAT3 consists of 770 amino acids that constitute six different functional domains, including an amino-terminal domain (NTD), coiled-coil domain (CCD), DNA-binding domain (DBD), linker domain, SH2 domain, and the carboxyl-terminal transactivation domain (TAD) [2]. In normal healthy cells, STAT3 is tightly regulated to a maintain transiently active state that plays a key role in many biological processes including differentiation, survival, proliferation, and angiogenesis [3]. However, aberrant or constitutively activated STAT3 has been observed in many different cancers, such as colorectal, lung, breast, prostate, liver, and pancreas cancers, and STAT3 activation associates with poor prognosis of these cancers [4,5,6,7,8,9]. Chronic inflammation is regarded as an important risk factor for the development of colorectal cancer, and IL-6/STAT3 activation seems to take a center stage in human cancer development [10,11]. The abnormal activation of STAT3 upregulates its downstream target genes encoding anti-apoptosis factors, cell-cycle regulators, metastasis, inducers of tumor angiogenesis, etc., [12,13], which are associated with the induction of cancer and the processes promoting the survival of cancer [14,15,16]. Thus, STAT3 has been considered to be an oncogene owing to its ability to promote malignancy through multiple processes [14,17].

Given the oncogenic functions of STAT3, directly targeting STAT3 signaling represents a potential therapeutic approach to treating cancer [8,18,19,20,21,22]. Dominant-negative STAT3 [23,24], RNA interference [25,26,27], STAT3 antisense oligonucleotides [28,29], phosphopeptide inhibitors [30,31,32], and small molecule inhibitors [33,34,35,36,37,38,39] have successfully resulted in an inhibition of growth and the induction of death in tumors, and several small-molecule STAT3 inhibitors have been reported in phase 1 clinical trials [40,41]. Moreover, it was also determined that in normal cells, blocking STAT3 is neither harmful nor toxic to the cells [19]. However, so far, no inhibitor of STAT3 has entered clinical application due to poor pharmacokinetic properties and peripheral nervous system toxicity from compound properties [22,42]. In addition, recent success of selective degradation for STAT3 protein has been made via proteolysis targeting chimera (PROTAC) technology that used a small molecule inhibitor of STAT3 as a ligand for target protein binding [43]. These reports all urge people to search for more effective STAT3 inhibitors.

In this study, we identified trichothecin (TCN) as a novel inhibitor of STAT3. TCN is a trichothecene mycotoxin isolated from marine endophytic fungi. Previous reports have shown that TCN has wide biological activities including anticancer, etc. [44,45,46]. However, the mechanism underlying anticancer effect of TCN is still not very clear. We demonstrate that TCN treatment inhibits the phosphorylation of STAT3 by directly binding to its SH2 domain, resulting in inhibition of dimerization, nuclear translocation, and transcriptional activity of STAT3, thereby inhibiting multiple cancer-related features in colorectal cancer development in vitro.

## 2. Results

### 2.1. TCN Induces Cell Apoptosis of Human Colorectal Cancer HCT 116 Cells

TCN has a tetracyclic 12,13-epoxy skeleton structure, the structure of which is shown in Figure 1A. In cell proliferation assays, TCN exhibited strong inhibitory activities on various cancer cell lines. Human colon cancer cell line HCT 116 was most sensitive to TCN with an IC_50_ value of 0.25 μM after 72 h treatment. It was noteworthy that the cytotoxic effects of TCN was significantly less pronounced on human normal live cells L-02 (IC_50_: 4.7 μM) (Figure 1B). In HCT 116 cells, TCN displayed proliferation inhibitory activities in a concentration- and time-dependent manner (Figure 1C). Furthermore, TCN also showed significant inhibitory effects on the plate colony-forming abilities of HCT 116 cells (Figure 1D,E). Hence, the HCT 116 cell line was chosen for further study of TCN. 

In order to explore whether the cell cycle arrest contributed to TCN-induced proliferation inhibition, we further analyzed the cell cycle distribution and found that TCN induced G0/G1 phase arrest in HCT 116 cells in a concentration-dependent manner (Figure 1F,G). Then, we determined whether G0/G1 phase arrest evoked by TCN resulted in cell apoptosis. Cysteinyl aspartate specific proteinase-9 (caspase-9) is the apical caspase in the intrinsic apoptosis pathways, and cysteinyl aspartate specific proteinase-3 (caspase-3) is considered to be the most important of the effector caspases. Cleaving and activation of caspase-9/caspase-3 is a hallmark of the intrinsic apoptosis [47,48]. In addition, poly-adenosine diphosphate-ribose polymerase (PARP) protein is a nuclear enzyme. Cleaved PARP seems to be an early marker of apoptosis in cells [49]. As shown in Figure 1I, after TCN treatment, caspase-3, caspase-9, and PARP were distinctly decreased in a concentration-dependent manner, and the cleaved PARP and caspase-9 increased correspondingly, suggesting that these proteins were cleaved after treatment of TCN. In addition, Bcl-2 (an apoptosis inhibitor) was downregulated, accompanied by a dramatic enhancement of Bax (an apoptosis promoter) after TCN treatment for 24 h. In order to quantify the extent of apoptosis, flow cytometry analysis was performed to evaluate the TCN-induced apoptosis rate by Annexin V-FITC and PI staining. After treatment with TCN (0, 0.25, 0.50, 1 μM) for 24 h, the rates of early apoptosis (AV^+^/PI^-^) were 8.36%, 20.50%, 27.32%, and 34.25% respectively (Figure 1H). These results indicate that TCN is able to promote cell apoptosis in colorectal cancer HCT 116 cells.

### 2.2. TCN Preferentially Inhibits Activation of STAT3 

To investigate the mechanism underlying proliferation inhibition of TCN, we first detected the expression and phosphorylation levels of STAT3, AKT, and ERK, which are molecules in the main three signaling pathways related to cell proliferation and survival [50]. After treatment with TCN (0–1 μM) for 24 h, the expressions of STAT3, AKT, ERK made no significant change on HCT 116 cells, while the phosphorylation levels of STAT3, AKT, ERK were obviously decreased. Specially, TCN displayed significantly decreased phosphorylation levels of STAT3 at a lower concentration (0.12 μM), while the inhibition effects on AKT and ERK phosphorylation were obvious until the TCN concentration reached 0.5 μM (Figure 2A). Similarly, we found that the phosphorylation level of STAT3 was decreased after treatment with TCN (0.5 μM) for 1 h, while that of ERK, AKT, and P65 only emerged after 3 h (Figure 2B). These results suggest TCN may mainly regulate STAT3 activation to inhibit proliferation of HCT 116 cells.

The activation of STAT3 was regulated by multiple cytokine and growth factor-mediated signaling pathways. The STAT3 pathway acts as a therapeutic target in head and neck cancers [22,51], and we further detected the expression and phosphorylation levels of multiple signal molecules upstream of STAT3. The results are shown in Figure 2C. After treatment with TCN (0.5 μM) for 1 h on HCT 116 cells, the phosphorylation of STAT3 (Y705) was dramatically decreased, while the phosphorylation levels of Src, gp130, JAK1, JAK2, and EGFR had no marked change. The phosphorylation levels of JAK1, JAK2, EGFR, and PDGFR were decreased only when the incubation time between 3 and 6 h. In addition, analysis of tyrosine kinase activity in vitro also showed that TCN had no obvious inhibitory effects on multiple tyrosine kinase receptors binding to various growth factor ligands (Table 1). All together, these results indicate that TCN may directly inhibit activation of STAT3 independent of any upstreaming signaling alterations. We noted that activation of STAT1 was not changed after TCN treatment (Figure 2D). Results generated from most STAT1 studies support the concept of STAT1 as a tumor suppressor in various cancers including colorectal cancer, which fulfills an opposite role to that of STAT3 [52,53]. Similar to STAT3, STAT5B has been mostly reported to play a major role in the progression and pathogenesis of cancer [54], but expression and phosphorylation levels of STAT5B were not affected by TCN. The results suggest that TCN exhibits a selectivity to target STAT3 to some extent.

### 2.3. TCN Inhibits the Dimerization of STAT3 and Blocks STAT3 Nuclear Translocation 

It is well known that when cytokines or growth factors specifically bind to their respective receptors, STAT3 is activated by the phosphorylation of its tyrosine residue (Y705), and then forms dimers and translocates from the cytoplasm into the nucleus to exert its function as a transcription factor [55,56]. Increased IL-6 expression has been related to advanced stages of disease and decreased survival in colorectal cancer patients, it activates JAKs with a subsequent activation of STAT3 through phosphorylation [57]. To determine whether TCN suppressed the dimerization of STAT3 and its biological function, HCT 116 cells were cultured in serum-free medium for 24 h and then pre-incubated with TCN (0.12–0.50 μM) for 90 min, followed by stimulation with IL-6 for 30 min. The level of STAT3 dimerization was found to increase in the presence of IL-6 as detected by Western blotting after cells were crosslinked with Bis(sulfosuccinimidyl)suberate (BS3, Thermo Fisher Scientific, Waltham, MA, USA). The effect was remarkably reversed by TCN treatment in a dose-dependent manner. (Figure 3A,B).

As mentioned above, TCN inhibited STAT3 activation and dimerization, we then hypothesized that TCN was able to prevent STAT3 from translocating from the cytoplasm to the nucleus. To test this hypothesis, HCT 116 cells were treated with TCN for 4 h. After fixation, immunofluorescence staining was conducted as described under “Materials and Methods”. As shown in Figure 3C, STAT3 was found to partially localize to the nucleus in HCT 116 cells although without IL-6 stimulation, while STAT3 was less retained in the nucleus when cells were treated with TCN compared with control, suggesting TCN treatment blocks nuclear translocation of STAT3.

Activated STATs enter the nucleus and bind to specific enhancer sequences in target genes, thus regulating their transcription [58,59]. Next, we detected the effect of TCN on transcriptional activity of STAT3. HeLa cells with stably transfected report gene plasmid pSTAT3-TA-luc were cultured with TCN for 4 h, and then luciferase activity and cell viability were detected. TCN (0.06–4 μM) treatment for 4 h had no cytotoxicity on HeLa (STAT3) cells (Figure 3D), but obviously decreased the detection signal of chemiluminescence in a concentration-dependent manner (Figure 3E). The IC_50_ value was 0.64 μM, showing that TCN inhibits the transcriptional activity of STAT3. Furthermore, TCN treatment on HCT 116 cells for 24 h downregulated the expression of STAT3 target genes, such as survivin, CyclinD1, Mcl-1, C-myc, MMP9, and VEGF, markedly and in a concentration-dependent manner (Figure 3F). These results indicate that TCN inhibits the dimerization of STAT3, resulting in impairment of STAT3 nuclear translocation and subsequent activity of transcription.

### 2.4. TCN Binds to the SH2 Domain of STAT3

TCN inhibited STAT3 activation and dimerization, blocked STAT3 nuclear translocation and transcriptional activity, and preferentially inhibited activation of STAT3 by means other than its upstream regulators on HCT 116 cells. We hypothesized that TCN may interact with STAT3 directly. To test this hypothesis, the molecular modeling was first carried out using the Fang Zheng wen Xiang E520 workstation SYBYL X 1.10 program package. The docking results showed that TCN bound to the SH2 domain through hydrophobic interactions, and formed hydrogen bonds with ASN538 and TRP501 (shown in brackets) located on other STAT3s in the dimerization of STAT3 through two carbonyls (Figure 4A). It is reported that the two inhibitors of STAT3, stattic and tyrosine phosphorylated oligo peptide GpYLPQTV-NH2, bind to the SH2 domain of STAT3 and inhibit STAT3 activation and dimerization [31,60]. In order to confirm that TCN binds to the SH2 domain of STAT3, we further detected the combination effects of TCN and stattic on the cytotoxicity in HCT 116 cells, and the competitive inhibition activity of TCN on GpYLPQTV-NH2 binding to STAT3. We found that TCN antagonized the cytotoxicity effect of stattic with a combination index (CI) greater than 1 (Figure 4B), which indicates TCN competes with stattic. By fluorescence polarization assays, 5-carboxyfluorescein-GpYLPQTV-NH2 linking with STAT3 generated a strong fluorescence polarization detection signal, while TCN attenuated the signal in a dose-dependent manner; the inhibition rates treated by TCN (0.4, 2, 10 μM) were 12.8%, 53.9%, 72.2%, respectively (Figure 4C). These results suggest that TCN binds to STAT3 with the binding site at the SH2 domain.

### 2.5. TCN Inhibits Cell Metastasis and Angiogenesis In Vitro

We have shown that TCN effectively inhibited STAT3 activation, suppressed its downstream target gene expression, and induced apoptosis in human colorectal cancer HCT 116 cells. We next examined what effect TCN’s ability to inhibit STAT3 would have on colony formation of HCT 116 cells in soft agar and on cell migration and invasion, which are important processes in tumor metastasis. As shown in Figure 5, colony growth in soft-agar suspension of HCT 116 cells was significantly inhibited by TCN compared with the DMSO control (Figure 5A); the colony formation rates of cells treated by TCN (0.06, 0.12, 0.25 μM) were 41.5%, 24.4%, and 6.9%, respectively, versus the DMSO control (Figure 5B). HCT 116 cells migrated within 24 h to the scratched area, and TCN (0.03, 0.06, 0.12 μM) treatment markedly prevented this effect; the relative migration distances to control were 48.6%, 51.8%, and 57.0% (Figure 5C,D). Similarly, HCT 116 cells invaded the matrigel to the lower side of the basement membrane, which was significantly antagonized by TCN treatment (Figure 5E,F). Furthermore, TCN decreased the expression levels of C-myc and MMP9, which are regulated by STAT3 and relate to a cell’s malignant transformation and metastasis (Figure 3G). It was noteworthy that TCN markedly inhibited colony formation, migration, and invasion of HCT 116 cells, mediated by IL-6 (Figure 6A–F), in a concentration-dependent manner.

Increasing evidence suggests that STAT3 is also an important facilitator of angiogenesis under both physiological and pathological situations [4,61]. To study the anti-angiogenic activity of TCN, a three-dimensional angiogenesis assay was performed. Human HUVEC cells were cultured onto matrigel for 18 h, and robust, elongated tube-like structures were formed. The number of formed tubules was counted using inverted phase contrast microscopy, which directly revealed the ability of the HUVECs to form tubular structures. As expected, TCN clearly inhibited HUVEC tube formation (Figure 5G,H). Furthermore, TCN also abolished the tube formation and plate colony-forming abilities of HUVEC cells induced by IL-6 (Figure 6G–J).

## 3. Discussion

It has been widely accepted that constitutive STAT3 activation provides tumor cells with a survival advantage. STAT3 has proved to be a promising molecule target in treating cancers [9,22,62,63]. Although there are currently no approved drugs targeting STAT3, many related efforts are still underway. In the present study, we demonstrate that TCN is a novel inhibitor of STAT3 by directly binding to the SH2 domain.

Constitutive activation of STAT3 in cancers results from activation of several oncogenic pathways. Most frequently, gain-of-function mutations and overexpression of tyrosine kinases, such as PDGFR, EGFR, FGFR, Her2/Neu, IL-6R/gp130, c-Met, and SRC, lead to the persistent activation of STAT. TCN was found to preferentially inhibit activation of STAT3 at a lower dosage and shorter time, and had no obvious inhibitory effects on multiple receptor tyrosine kinases. The main domains of the STAT3 molecule include the N-terminal regions for cooperative DNA binding of two STAT dimers, DNA binding domain, Src homologous region 2 (SH2) domain for dimerization, and the transcription activation domain at the C-terminal end region [2]. Over the last 20 years, various small molecule inhibitors targeting different domains of STAT3 have been tested for discovery and development of potential therapeutics. However, the vast majority of past STAT3 inhibitors targeted the SH2 domain. TCN was revealed to be an inhibitor against the SH2 domain by competitive inhibition assay and molecular modeling. The SH2 region is the most conservative part of the STAT3 structure, which plays an important role in signal transduction by recognizing the specific SH2 binding domain of activated upstream receptors to aggregate in the direction of the receptor and activating JAKs family. STAT3 is then activated by phosphorylation at tyrosine residue 705 (Tyr-705), which leads to p-Tyr-705-SH2 dimer formation, nuclear translocation, recognition of STAT3-specific DNA binding elements, and activation of target gene transcription [59]. Consistently, TCN inhibited STAT3 activation (Tyr-705) and dimerization and blocked STAT3 nuclear translocation and transcriptional activity. It has been reported that mutations in the SH2 domain of STAT3, such as Y640F and D661Y in large granular lymphocyte leukemia, lead to constitutive activation of STAT3 by enhancing STAT3 protein phosphorylation and dysregulation of genes downstream of STAT3 [64]. TCN, by binding to the SH2 domain through hydrophobic bonds and forming hydrogen bonds with ASN1538 and TRP1501, may induce steric hindrances for SH2-upstream receptor binding and p-Tyr-705-SH2 dimer formation to inhibit constitutive activation of the mutant STAT3, which is worth studying in the future.

TCN has been reported before to have widely biological activities including inhibitions of protein synthesis [65,66,67], DNA and RNA synthesis, and mitochondrial function [68]. However, trichothecin inhibits around 90% of protein synthesis activity by around 90% at 100 μM (final concentration) [66]. We found TCN at a much lower concentration (0.12 μM) significantly abolished the phosphorylation level of STAT3 in HCT 116 cells, and no determined protein in expression level was downregulated, indicating that trichothecin exerts anticancer activity unrelated to the mechanisms suggested previously. In our study, we noted that the phosphorylation level of STAT3 was decreased after treated with TCN (0.5 μM) for 1 h, which was quick4r than that of P65. Jia Su reported that TCN works as a potent inhibitor of the NF-kB signaling pathway to induce cell death in NF-kB constitutively activated human cancer cells [69]. However, no precise target molecule of TCN was identified. Stark lab reported that overexpression of a constitutively active form of STAT3 in HME cells induced 427 genes, including TNF-α (a major NF-κB activator) [70]. Considering that NF-kB inhibitory activity was determined at 24 h or 48 h by Jia Su, we have reason to believe that TCN first targets STAT3 and then subsequently inhibits the NF-κB signaling pathway.

The dysregulation of STAT3 downstream target genes, including those involved in survival, proliferation, angiogenesis, and suppression of host immune surveillance, has been implicated in tumor initiation and formation [71,72]. In line with these reports, TCN exhibited strong proliferation inhibitory activity on various cancer cell lines. Specially, TCN induced cell apoptosis, migration, and invasion of human colorectal cancer HCT 116 cells, and tube formation of HUVEC. IL-6 is one of the abundantly expressed cytokines in the colorectal cancer microenvironment through gp130 activation on tumor cells with subsequent signaling through Janus kinases (JAKs) and signal transducers and activation of transcription 3 (STAT3). TCN was found to inhibit colony formation, migration, and invasion of HCT 116 cells mediated by IL-6. 

Our findings indicate that TCN, as a potent inhibitor of STAT3, has promising therapeutic value for cancer treatment and deserves further exploration.

## 4. Materials and Methods 

### 4.1. Compound

Trichothecin (TCN) was provided by Qianqun Gu group, School of Medicine and Pharmacy, Ocean University of China. It was isolated from a sponge-derived fungus *Alternaria* sp. TZP-11. TCN was solubilized in sterile dimethylsulfoxide (DMSO, Solarbio, Beijing, China).

### 4.2. Cell and Cell Culture

Human lung cancer cell lines A549, NCI-H1975, Hcc827, human colon cancer lines HCT 116 and HT-29, human breast cancer cell lines MDB-MA-231 and MCF-7, human gastric cancer cell lines MGC80-3 and MKN-28, human cervical cancer cell lines HeLa and SiHaand Caski, human ovarian cancer cell line HO-8910, human hepatocacinoma cell line BEL-7402, human prostate cancer cell line PC-3, human leukemia cell lines K562 and HL-60, and human normal liver cell line L-02 were purchased from the Institute of Biochemistry and Cell Biology, Chinese Academy of Sciences (Shanghai, China). The HUVEC cell line was purchased from the American Type Culture Collection (ATCC, Manassas, VA, USA). These cells were cultured in DMEM (HCT 116, HT-29), RPMI 1640 (NCI-H1975, Hcc827, MGC80-3, MKN-28, HO-8910, BEL-7402, PC-3, K562, HL-60, L-02, HUVEC), F12K(A549), and MEM (MCF-7, HeLa, SiHa, Caski,) medium, containing 100 U/mL penicillin, 100 mg/mL streptomycin, and 10% fetal bovine serum, in a humidified atmosphere of 5% CO_2_ at 37.0 °C.

### 4.3. Cell Viability Assay

Cell viability was measured by the SRB assay [73]. Briefly, cells were seeded onto 96-well plates (0.3 × 10^4^–0.7 × 10^4^ cells/well) and allowed to attach for 24 h, TCN was added to the medium at various concentrations for 72 h. Cold trichloroacetic acid (10%) was added and cells were fixed at 4 °C for 1 h, then washed with deionized water five times and dried in air. After that, SRB reagent (4 mg/mL, 100 μL/well, Sigma, Saint Louis, MO, USA) was added into the plates and the cells were incubated for 30 min at room temperature. After being washed with glacial acetic acid (1%) and dried in the air, the SRB reagent was dissolved in Tris-base (pH 8.0) 150 μL per well. The absorption value was detected by a microplate reader (BioTek, Winooski, VT, USA) at 515 nm. The cytotoxicity of TCN was expressed as an IC_50_ value.

### 4.4. Clonogenic Assay

HCT 116 cells were seeded at 150 cells/well in 6-well plates. Cells were treated with 0.06–0.25 μM TCN for 10 days, then were fixed by methanol at room temperature and stained with Giemsa solution. Colonies were defined as >10 cells/colony. Finally, colonies were counted and photographed.

### 4.5. Cell Cycle Analysis

HCT 116 cells were incubated with TCN (0–0.5 μM) for 24h. After that, the cells were collected and washed in PBS and fixed in ice-cold 70% (*v*/*v*) ethanol overnight at –20 °C. The cell pellet was resuspended in PBS and stained with a mixture of RNase (10 μg/mL) and PI (50 μg/mL) on sodium citrate containing 0.5% Triton X-100 for 20 min in the dark. Cell cycle distribution analysis was performed using an Aria FACS flow cytometry system (Beckman Coulter, Boulevard Brea, CA, USA).

### 4.6. Annexin V-FITC/PI Double-Staining Assay

The early apoptosis was performed using the Annexin V-FITC/PI apoptosis detection kit. Briefly, cells (3 × 10^5^) were incubated with TCN (0–1.0 μM) for 24 h, and then harvested by centrifugation, washed with ice-cold PBS twice, and resuspended in binding buffer. Staining was started by adding Annexin V-FITC (5 μL) and PI (5 μL) followed by incubated for 10 min at room temperature in the dark. Then, samples were immediately analyzed by an Aria FACS flow cytometry system (Beckman Coulter, Boulevard Brea, CA, USA).

### 4.7. Soft Agar Colony Formation Assay

HCT 116 cell colony forming activity was assayed in six-well plates using low gelling temperature agarose, as described previously [74]. Briefly, 2 mL of 0.6% bottom agarose layer in complete culture medium was poured into each well. After the hardening of the bottom layer, 3 × 10^3^ cells mixed in culture medium containing 0.35% agarose and different concentrations of TCN, were added to the plates. Two weeks later, the colonies were stained with crystal violet (0.005%) for 1 h in an incubator at 37 °C and 5% CO_2_. Colonies were defined as >50 cells/colony. Finally, colonies were counted and photographed.

### 4.8. Wound Healing Assay

Cells were seeded in 96-well plates and allowed to attach for 24 h. The cellular monolayer was wounded with a sterile 10 μL pipette tip and washed three times with serum-free DMEM to remove detached cells from the plates. The cells were incubated in the presence or absence of TCN for 24 h in serum free DMEM, and then were photographed using a phase-contrast microscope under a 4 × objective. The migration distance was assayed by ImageJ software.

### 4.9. Transwell Invasion Assay

Cell invasion assays were carried out with Corning Transwell Invasion Chambers with inserts. Each insert was 6.4 mm in diameter and had a 8.0 μm pore size and a PET porous membrane that was coated with matrigel matrix (BD Bioscience, Bedford, MA, USA). Cells (2.5 × 10^4^) were suspended in 200 μL of serum-free DMEM with or without TCN and were added to the upper chamber. Then, 600 μL of DMEM containing 20% FBS was added to the lower chamber of the 24-well plates. After incubation for 24h, non-invading cells were removed from the upper surface of the membrane by scrubbing, and invading cells on the lower surface of the membrane were stained with crystal violet. The invading cells were counted randomly by light microscopy under a 20 × objective. Each assay was repeated at least three times. Due to variation in the number of migrated cells from different experiments, the results were normalized to control cells and the relative invasion is expressed as mean ± SD of invasion cells relative to control cells.

### 4.10. HUVEC Tube Formation Assay

Cell culture plates (96-well) were bottom-coated with a thin layer of matrigel (50 μL/well), which was left to polymerize at 37 °C for 60 min. HUVEC cells (2–3 × 10^4^ cells) were added to each well on the solidified matrigel. Culture medium was added to each well in the presence or absence of TCN (0.12–0.50 μM). The plates were incubated at 37 °C for 18 h and the endothelial tubes were photographed. Tube forming ability was quantified by counting the total number of cell clusters and branches under a 4 × objective and four different fields per well. The results are expressed as mean ± SD of branching compared with the control group. Each experiment was performed at least three times.

### 4.11. Molecular Modeling

Molecular modeling was performed on a Fang Zheng wen Xiang E520 workstation using the software package of SYBYL X 1.10. The crystal structure of a STAT3 complex (PDB ID:3CWG) [75] was used as a structural template for docking studies. The 3D structural models of ligands (STAT3 and TCN) were generated using SurFlex modular [76].

### 4.12. Luciferase Report Gene Assay

Plasmid pSTAT3-TA-luc (Beyotime Biotechnology, Shanghai, China) contains multiple copies of the STAT3-binding site at its polyclonal site and its activation specifically depends on STAT3 status in the cell environment. pSTAT3-TA-luc plasmids were transfected into HeLa cell lines using Lipofectamine 2000 reagent (Invitrogen, Carlsbad, CA, USA). The stable clones, which showed high luciferase activity, were seeded into 96-well plates and cultured with or without TCN for 4 h, then, luciferase activity was measured by using a Steady-Glo^®^ Luciferase Assay System (Promega, Madison, WI, USA), following the manufacturer’s instructions. The cell viability was detected by SRB assay.

### 4.13. Immunofluorescence Assay

HCT 116 cells were seeded on glass coverslips in a 6-well plate. The next day, the cells were treated with TCN for 4h. Cells were fixed with cold paraformaldehyde (4%) for 30 min, permeated with 0.1% Triton X-100 for 15 min, and blocked with 1% BSA for 1 h. The cells were incubated with primary antibodies of STAT3 (1:100, CST, Boston, MA, USA) overnight at 4 °C. After incubation with FITC-labeled goat antirabbit IgG (H + L) for 30 min at room temperature, the cells nuclei were stained by incubation with 300 nM DAPI (Beyotime Biotechnology, Shanghai, China) for 5 min at room temperature. After mounting on a coverslip, 1024 × 1024 pixel images were taken using a Zeiss LSM510 confocal microscope. Image processing was performed using the Zeiss Image Browser software package.

### 4.14. Fluorescence Polarization Assays

The fluorescence polarization assays were performed at approximately 37 °C, as described previously [77]. The final concentration of buffer components used for all fluorescence polarization (FP) assays was 10 mM HEPES (pH7.5), 1mM EDTA, 0.1% Nonidet P-40, 50mM NaCl, and 10% DMSO. The sequence of the peptide was 5-carboxyfluorescein-GpYLPQTV-NH_2_ (where pY represents phospho-Tyr). Peptides were >95% pure. STAT3 proteins (final concentration: 210 nM, ab43618, Abcam) were incubated with TCN in Eppendorf tubes at 37 °C for 1h prior to addition of the 5-carboxyfluorescein labeled peptides (final concentration: 10 nM) Before measurement at room temperature, the mixtures were allowed to equilibrate for at least 30 min. Test compounds were used at the indicated concentrations diluted from a 20× stock in DMSO. The fluorescence polarization signals were detected by the Cytation 5 Hybrid Multi-Mode Reader (Biotek, Winooski, VT, USA). The inhibition rate (%) was calculated using the following equation: (1-(FP/FP control)) × 100%. Each assay was performed at least three times. The results are expressed as means ± SD

### 4.15. Tyrosine Kinase Activity Assay

The effects of TCN on the activities of various tyrosine kinases were determined using ELISA assays [78]. Briefly, 20 μg/mL poly (Glu, Tyr) 4:1 (Sigma, St Louis, MO, USA) was pre-coated in 96-well plates as a substrate. Then, a 50 μL aliquot of 10 μM ATP solution diluted in kinase reaction buffer was added to each well; 1 μL of various concentrations of TCN were then added to each reaction well. The kinase reaction was initiated by the addition of purified tyrosine kinase proteins diluted in 49 μL of kinase reaction buffer. After incubation for 1 h at 37 °C, the plate was washed three times with PBS containing 0.1% Tween 20 (T-PBS). Anti-phosphotyrosine (pY99) antibody (1:500) was then added. After incubation for 30 min at 37 °C, the plate was washed three times, and 100 μL horseradish peroxidase-conjugated goat antimouse IgG (1:2000) was added. The plate was then incubated at 37 °C for 30 min and washed three times. Then, 100 μL aliquot of a solution containing 0.03% H_2_O_2_ and 2 mg/mL o-phenylenediamine in 0.1 M citrate buffer (pH 5.5) was added. The reaction was terminated by the addition of 50 μL of 2 M H_2_SO_4_ as the color changed, and the plate was analyzed using an Epoch 2 Microplate reader (Biotek, Winooski, VT, USA) at 490 nm. The inhibition rate (%) was calculated using the following equation: (1-(A490/A490 control)) × 100%.

### 4.16. Western Blotting Analysis

Cells were incubated with various concentrations of TCN for appropriate times and then washed twice in cold phosphate buffered saline (PBS). Cells were lysed with lysis buffer (10mM Tris, pH 7.4, 150 mM NaCl, 1 mM ethylenediaminetetraacetic acid (EDTA), 1% Triton X-100, 0.5% NP-40, 1 mM propidium iodide (PI), 1mM dithiothreitol (DTT), 1mM phenylmethylsulfonyl fluoride (PMSF)) and placed on ice for 1h with occasional vortexing. Centrifugation followed at 10,000 rpm for 10 min and each cell lysate (50 μg) was subjected to sodium dodecyl sulfate (SDS)-polyacrylamide gel electrophoresis (PAGE) and transferred to nitrocellulose membranes (Pall, New York, NA, USA). Blots were blocked with 5% skim milk in TBST for 1h at room temperature, then incubated with indicated primary antibodies overnight at 4 °C, followed by incubation with antirabbit or antimouse horseradish peroxidase-conjugated immunoglobulin G (IgG) and visualized with enhanced chemiluminescence.

### 4.17. Statistical Analysis

Data in graphs are presented as mean ± SD. Statistical significance was analyzed by one-way ANOVA for at least three independent experiments, and *p* < 0.05 was considered significant.

## Figures and Tables

**Figure 1 molecules-25-02306-f001:**
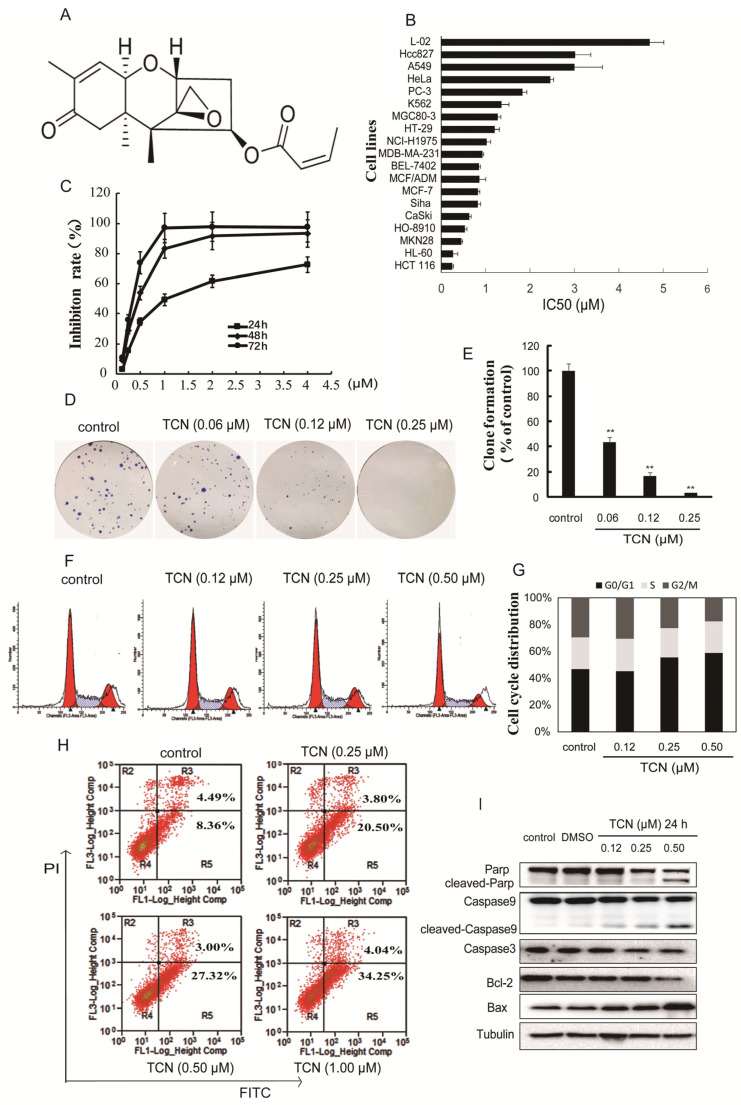
Cytotoxicity of trichothecin (TCN) and the effects of TCN on colony formation and apoptosis in HCT 116 cells. (**A**), Chemical structure of trichothecin (TCN). (**B**(, The cytotoxicity of TCN was evaluated on various cell lines. Cells were seeded in 96-well plates, then treated with TCN for 72 h, and the cell viability was assayed by Sulforhodamin B (SRB) assay. The cytotoxicity of TCN was expressed as IC_50_. (**C**), TCN inhibited proliferation of HCT 116 cells in a time- and concentration-dependent manner. (**D**), Inhibitory effect of TCN on the colony formation of HCT 116 cells. Cells were treated with TCN (0–0.25 μM) for 10 days and stained with Giemsa. Then photos were taken and counted. (**E**), Quantification of the number of colonies. Colonies consisting of ≥50 cells were scored. Data are presented as means ± SD from three independent experiments. ** *p* < 0.01 vs. control. (**F**), TCN affected cell cycle distribution of HCT 116 cells. Cells were treated with TCN (0–0.50 μM) for 24 h. Cells were collected, digested with RNase A, and stained by PI. The DNA content of the cells was determined with the Aria FACS flow cytometry system. (**G**), Histograms show the percentage of cells in G0/G1, G2/M, and S phase after treatment with TCN. (**H**), Apoptosis rates in HCT 116 cells induced by TCN. HCT 116 cells were treated with indicated concentrations of TCN for 24 h, respectively, stained with Annexin V-FITC/PI, and determined by flow cytometry. R2 represents the necrosis cells, R3 represents the late apoptosis cells, R4 represents the normal cells, and R5 represents the early apoptosis cells. (**I**), Effects of TCN on apoptosis-related proteins. HCT 116 cells were treated with indicated concentrations of TCN for 24 h, and the levels of apoptosis-related proteins were detected by Western blotting. Tubulin served as the loading control.

**Figure 2 molecules-25-02306-f002:**
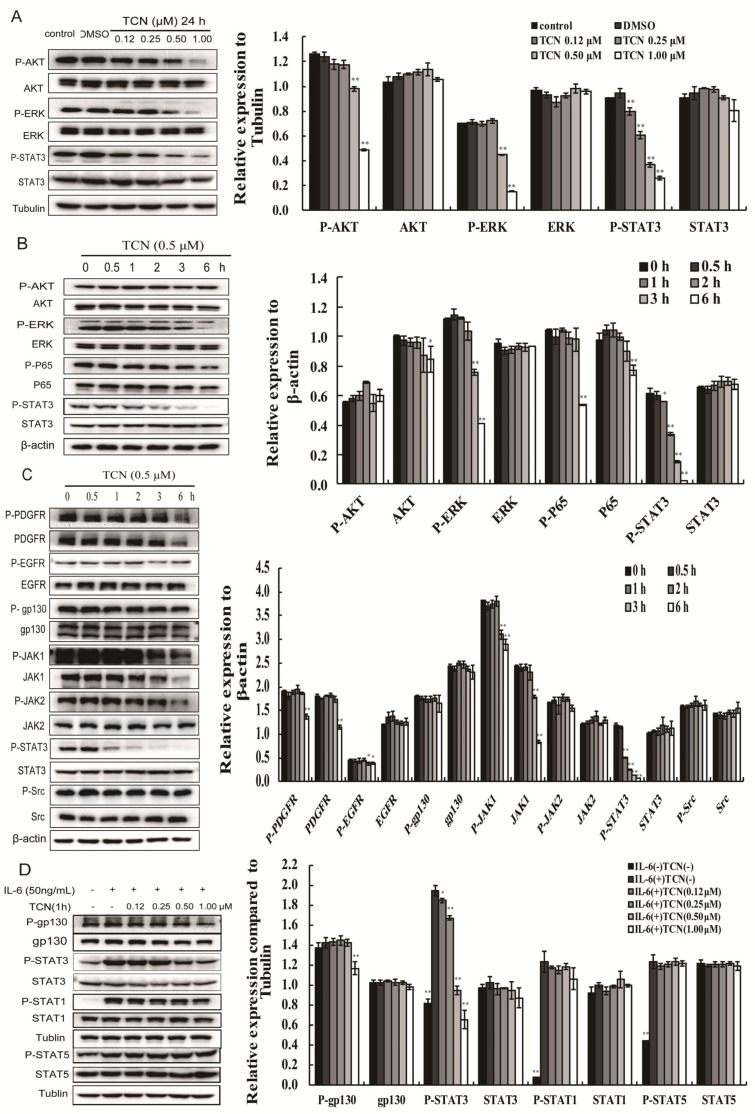
TCN inhibits STAT3 but not its upstream regulators on HCT 116 cells. (**A**), HCT 116 cells were treated with TCN (0–1.00 μM) for 24 h, then lysed and assayed by Western blotting. Tubulin served as the loading control. Protein band densities were quantified by normalizing to tubulin. * *p* < 0.05, ** *p* < 0.01 vs. control. (**B**,**C**), HCT 116 cells were treated with TCN (0.5 μM) for 0, 0.5, 1, 2, 3, and 6 h, then lysed and assayed by Western blotting. β-actin served as the loading control. Protein band densities were quantified by normalizing to β-actin. * *p* < 0.05, ** *p* < 0.01 vs. 0h. (**D**), HCT 116 cells were treated with TCN (0–1.00 μM) for 30 min, before IL-6 stimulation for 30 min, then lysed and assayed by Western blotting. Tublulin served as the loading control. Protein band densities were quantified by normalizing to tubulin. * *p* < 0.05, ** *p* < 0.01 vs. IL-6(+)TCN(-) group.

**Figure 3 molecules-25-02306-f003:**
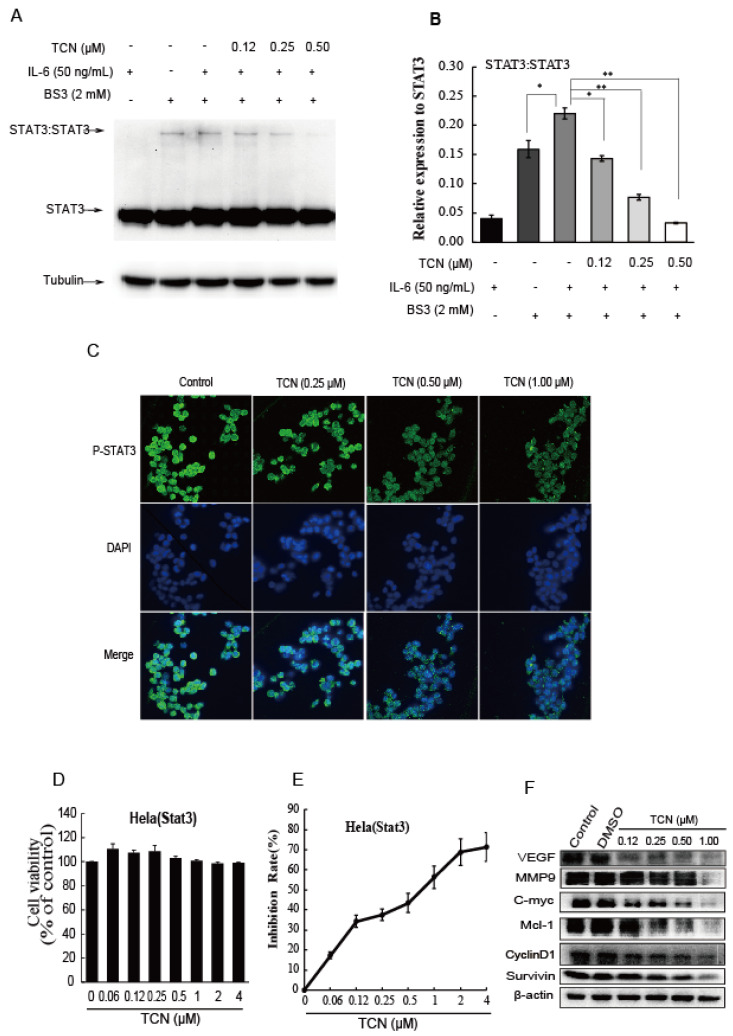
TCN inhibits the dimerization of STAT3 and blocks STAT3 nuclear translocation. (**A**), Dimerization analysis. HCT116 cells were preincubated with TCN for 90 min, then treated with IL-6 for 30 min. After cross-linking the proteins by adding BS3, the cells were lysed and detected by Western blotting. Tubulin served as the loading control. (**B**), Quantification analysis of A. Data are presented as means ± SD from three independent experiments. * *p* < 0.05, ** *p* < 0.01. (**C**), Immunofluorescence assay. HCT 116 cells were treated with TCN for 4 h, then incubated with P-STAT3(Y705) antibody overnight at 4 °C, then incubated with FITC-goat antirabbit antibody for 30 min at room temperature in the dark. The samples were assayed using laser confocal microscopy. Green represents P-STAT3, blue represents the nucleus. (**D**), HeLa cells transfected plasmid pSTAT3-TA-luc were seeded in 96-well plates, then treated with TCN (0–4 μM) for 4 h. The cell viability was detected by SRB assay. (**E**), Luciferase activity was measured by using a Promega luciferase kit. The fluorescence intensities (FI) were detected, and the inhibition rate (%) was calculated using the following equation: (1-(FI/FI control)) × 100%. (**F**), HCT 116 cells were treated with TCN (0–1.00 μM) for 24 h, the levels of VEGF, MMP9, C-myc, Mcl-1, CyclinD1, and survivin were assayed by Western blotting. β-actin served as the loading control.

**Figure 4 molecules-25-02306-f004:**
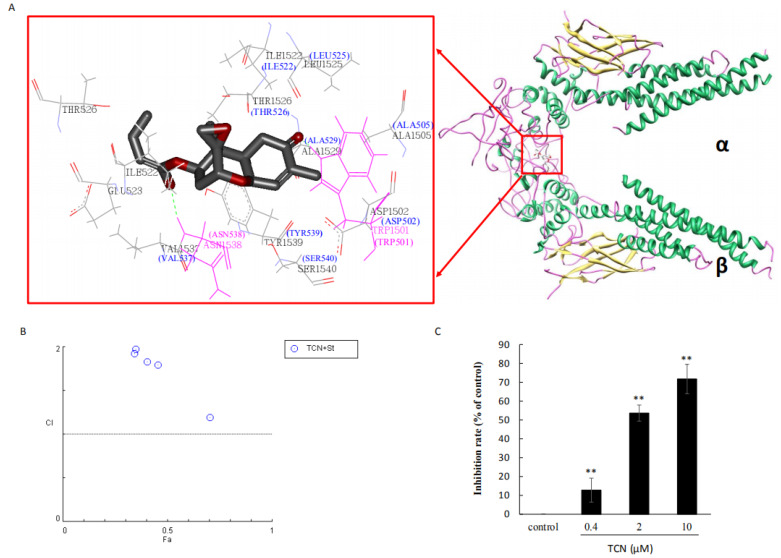
TCN binds to the SH2 domain of STAT3. (**A**), Docking model of TCN binding to the STAT3 SH2 domain (Protein Crystal Data Bank code 3CWG). Carbon atoms of TCN are colored gray, oxygen atoms of TCN are colored red. The extra “1” in front of the residue numbers is an artifact of the Protein Crystal Data Bank (PDB) when examining the second crystallographic dimer of STAT3 in 3CWG. The labels in brackets are actual residue numbers. (**B**), The combination index (CI) of TCN combined with stattic. The CI value was calculated according to Compusyn software. CI > 1 is antagonistic, CI = 1 is additive, and CI < 1 is synergistic. (**C**), Fluorescence polarization assay. STAT3 Proteins were incubated with TCN at 37 °C for 60 min prior to addition of the 5-carboxyfluorescein labeled peptides (final concentration: 10 nM). The fluorescence polarization (FP) signals were detected by Microplate Reader. The inhibition rate of the FP signal was calculated using the following equation: (1-(FP/FP control)) × 100%. Data are presented as means ± SD from three independent experiments. ** *p* < 0.01 vs. control.

**Figure 5 molecules-25-02306-f005:**
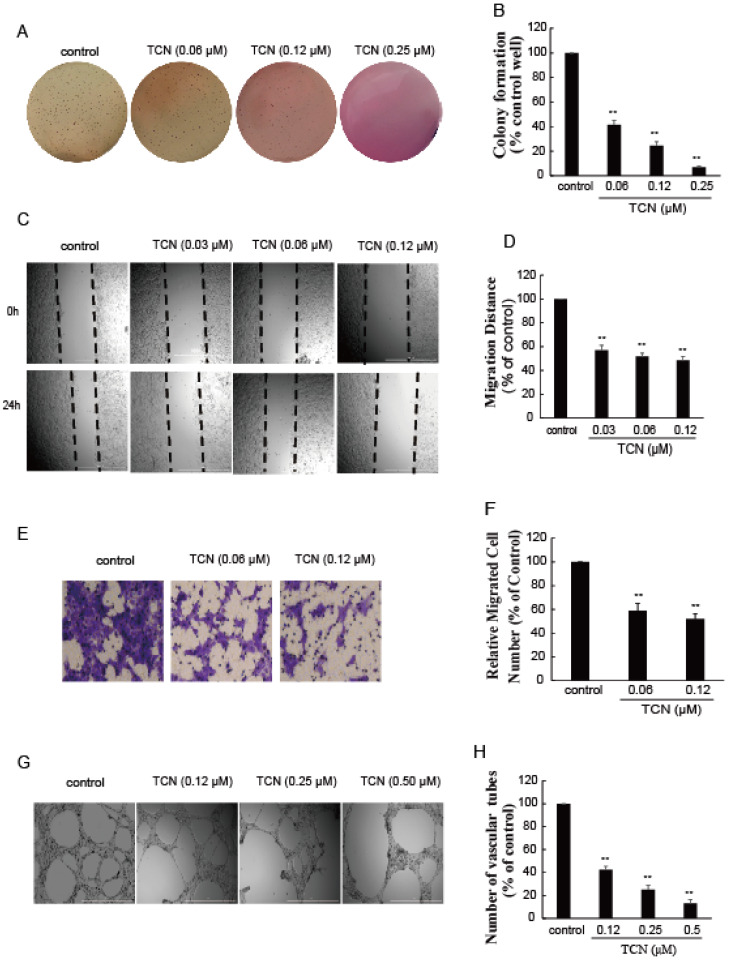
TCN inhibits the metastasis and angiogenesis on HCT 116 cells. (**A**), HCT 116 cells were mixed in completed culture mediums containing 0.35% agarose in the absence or presence of TCN. Two weeks later, the colonies were stained by crystal violet, then were photographed and numbered. (**B**), Quantitative analysis of the number of colonies in A. (**C**), HCT 116 Cells were treated with or without TCN for 24 h, cell migration was determined using a scratch-wound assay. (**D**), Quantitative analysis of cell migrations in (**C**). Results are normalized to control. (**E**), HCT 116 cells were seeded on transwell, then treated with or without TCN. After incubation for 48 h, the cells were stained by crystal violet and photographed. (**F**), Quantitative evaluation of cell invasion in (**E**). Results are normalized to control. (**G**), HUVEC cells were carefully layered on top of the polymerized matrigel, then treated with or without TCN. After 18 h, the photos were taken and vascular tubes were counted. (**H**), Quantitative evaluations of vascular tubes in G. Results are normalized to control. All experiments were repeated more than three times. Values represent the means ± SD of triplicate measurements. ** *p* < 0.01 vs. control.

**Figure 6 molecules-25-02306-f006:**
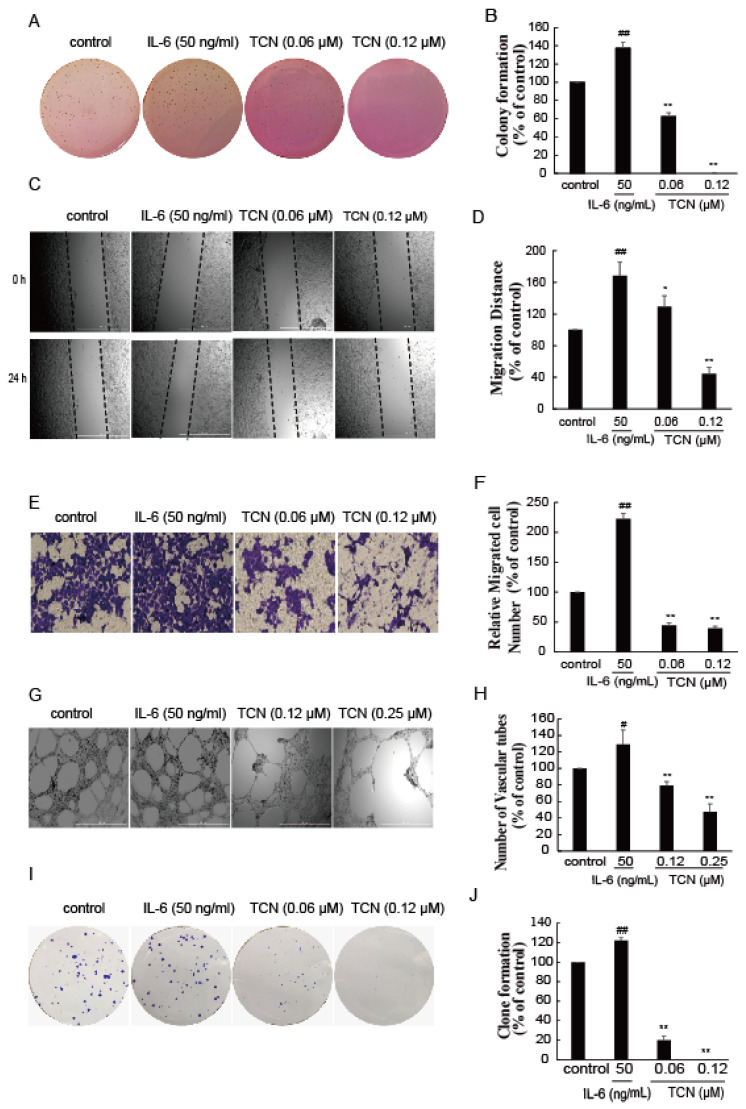
TCN inhibits the metastasis and angiogenesis on HCT 116 cells induced by IL-6. (**A**), HCT 116 cells were mixed in culture medium containing 0.35% agarose in the absence or presence TCN and IL-6 (50 ng/mL). Two weeks later, the colonies were stained by crystal violet, then were photographed and numbered. (**B**), Quantitative analysis of the number of colonies in A. (**C**), HCT 116 cells were treated with or without TCN and IL-6 for 24 h, cell migration was determined using a scratch-wound assay. (**D**), Quantitative analysis of cell migrations in (**C**). Results are normalized to control. (**E**), HCT 116 cells were seeded on transwell, then treated with or without TCN and IL-6. After incubation for 48 h, the cells were stained by crystal violet and photographed. (**F**), Quantitative evaluations of cell invasion in E. Results are normalized to control. (**G**), HUVEC cells were carefully layered on top of the polymerized matrigel, then treated with or without TCN and IL-6. After 18 h, the photos were taken and vascular tubes were counted. (**H**), Quantitative evaluations of vascular tubes in G. Results are normalized to control. (**I**), HCT 116 cells were treated with or without TCN (0–0.25 μM) and IL-6 for 10 days and were stained with Giemsa. Then, photos were taken. (**J**), Quantitative analysis of the number of colonies in I. Results are normalized to control. All experiments were repeated more than three times. Values represent the means ± SD of triplicate measurements. ^#^
*p* < 0.05, ^##^
*p* < 0.01 vs. control, * *p* < 0.05, ** *p* < 0.01 vs. IL-6 group.

**Table 1 molecules-25-02306-t001:** Inhibition rates of TCN on various tyrosine kinase activities.

Compounds	Concentration (μM)	Inhibition Rate(%)									
		Flt-1	KDR	PDGFR-α	PDGFR-β	EGFR	ErbB4	c-Src	ABL	EPH-A2	IGF1R
TCN	20.0	0	16.4	23.1	1.0	11.6	21.4	0.0	0.0	0.0	0.0
Su11248	1.0	66.6	81.8	68.4	80.2						
BIBW2992	1.0					84.7	84.2				
Dasatinib	1.0							63.2	56.7	76.2	
AEW541	1.0										56.6

The effects of TCN on the activities of various tyrosine kinases were determined using ELISA assays. The inhibition rate (%) was calculated using the following equation: (1-(A490/A490 control)) × 100%. Su11248, BIBW2992, Dasatinib, and AEW541 were used as positive control.

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
