# Peer review of "Trichothecin Inhibits Cancer-Related Features in Colorectal Cancer Development by Targeting STAT3"

_molecules, 2020, doi:10.3390/molecules25102306_

Round 1

Reviewer 1 Report

This manuscript discusses the finding of a natural product, Trichothecin, as an inhibitor of STAT3. Phenotypic and cellular target engagement suggests selectivity for STAT3. The manuscript will be acceptable for publication, pending a few “major” revisions, as well as adjusting the manuscript with the minor suggestions.

Major Revisions:

  • The Figures are of extremely low quality in the .pdf version that is available for review. Figure 4 is barely legible, and the data cannot be reviewed in a meaningful way. This can likely be readily resolved by authors, but as it currently is shown, none of the figures are acceptable for publication based on the pixelated image quality.
  • The reference list should be updated, and some are missing. Software packages likely need to be referenced appropriately (usually there is an information package with the software to indicate which references need to be included). Although the PDB code is listed, the reference to the paper for the STAT3 crystal structure is missing. Some references need to altered, for instance in Line 53, the inhibitors listed are from 10-15 years ago. There are a few recent reviews published on STAT3 inhibitors as well as severall others that can be included (see Andrea Wong, et al Boon-Cher Goh in 2017 (Expert Opinion on Investigational Drugs) or Anna Orlova et al Richard Moriggl in 2019 (Cancers)).
  • Docking results with STAT1 (PDB: 1YVL) would be interesting to examine if there is a potential selectivity window between the two proteins. Also, inhibition of STAT3 is sometimes compensated by upregulation of STAT5B. Docking with STAT5B (PDB: 6MBW) would also be interesting to observe whether Trichothecin interacts with STAT5B in a similar manner to STAT3.

Minor Revisions:

Throughout the manuscript, there should be consistency between use of IUPAC labels (for example, hour vs hrs vs h) and spacing (10 mM and 10mM, 30℃ vs 30 ℃ etc.)

Title: Suggestion to remove the “against”

Affiliations: pending journal requirements, the affiliation of #3 is used before #2

Line 18: “has” to “is”

Line 22: level to levels

Line 52: spelling of successfully

Line 54: Change “no any inhibitor”

Line 57: reword “has made”

Line 62: has widely to has wide

Line 133: ligands

Line 133: inhibit

Line 148: lysed

Line 166: partially localize to the

Line 212: The last residue of STAT3 is 770, yet residues ASN1538 and TRP1501 are listed.

Line 214: Include references

Line 216: Spelling of Stattic

Line 220: Unless this is a misinterpretation of the figure based on the image quality, can the authors comment on the reason for including a non-conventional FP assay. Generally, FP assays with an inhibitor would be performed at a single concentration of peptide and varying concentrations of inhibitor to determine the IC50 for peptide displacement. This is how the assays were routinely performed in the recent Berg and Gunning Lab papers.

Line 233: spelling concertration

Figure 4: Delete extra Figure

Figure 6: inhibites to Inhibits

Line 293: “STAT3 has proved to promising” to “proved to be a” Also include some references

Line 302: concludes to includes

Line 302: dimerization domain at N-terminus, the N-terminus is generally for tetramerization/oligomerization

Line 308: homologous or heterodimer to homo- or hetero-dimer

Line 325: remove “by them”

Line 415: References for the software package likely need to be included.

Line 423: A reference should be provided for the nucleotide sequence.

Line 448: ‘describer’ to described

Line 448: Additional details should be provided on the protein used for the FP assays with additional references to the recent Berg/Gunning group STAT papers on the development of the peptide assays.

Line 451: phosphor to phospho

Line: 466: PY699 is conventionally referred to as pY699

Line 486: The first two words should probably be removed

Author Response

Please see the attachmen.

Reviewer 2 Report

The study describes the effects of TCN, a natural marine compound, on STAT3 signaling in multiple cell lines. The authors then focus their work on the HCT-116 cell line, which was shown to be the most sensitive to TCN.

TCN has been shown to moderate multiple different cellular pathways that could be upstream STAT3, but the authors make the distinction that decreases in STAT3 phosphorylation appear to precede changes in receptors such as PDFR and EGFR.  Later molecular studies show TCN to be binding in the SH2 domain STAT3 and this is followed by other functional tumorigenesis assays.

The impact of the study would have been higher if attempts to restore or rescue STAT3 signaling while TCN is present would have been attempted.  This would more clearly have shown that STAT3 signaling is critical to the effect in functional assays.  Compound derivatization and optimization in future studies might help untangle the multiple effects of TCN if they are shown to have varying activities against various SH2 domains.

Specific critiques:

In figure 3A, quantification of the STAT3 band should also be done as STAT3 phosphorylation relative to total STAT3 protein is a better metric of whether there is a specific inhibition. 

In figure 3D, the dose response reduction in DNA binding does not seem obvious.   If phosphorylation, dimerization and transit to the nucleus are all inhibited by TCN, it would not be necessary to block DNA binding in order to block STAT3 signaling.  How does TCN, which binds in the SH2 domain, cause problems with DNA binding as this is a separate domain? This mechanism should be discussed and contrasted to previous SH2 domain inhibitors.  It should be mentioned in the results that this activity required a higher concentration of TCN and may not be specific. 

It is also not speculated how TCN prevents dimerization, is this solely through the prevention of STAT3 Y705 phosphorylation?  Or would TCN block dimerization even in the presence of phospho Y705?  It should be made clearer in the discussion (line 313) that all effects seen are dependent on the lack of activation. Or not?

Figure 1I results are poorly described in the next.  It is not just the reduction in full length PARP or Caspase 9 that should be noted, but also the appearance of cleaved PARP and cleaved Caspase 9, which are apparent in the blot and should be noted.

In general, the resolution of figures in the manuscript did not allow for proper review and must be improved.  Figure 4 A-D was not at all readable.  There have been mutations reported in STAT3 in LGL leukemia, Y640F, D661Y and others.  It may make sense to mention how these might affect binding of the compound when describing 4B.

There are many, but quite small, English language errors that need to be corrected.

Reviewer 3 Report

The manuscript submitted by Qi et al. is really interesting and contains promising results for STAT3 pharmacological inhibition. However, several issues and concerns must be addressed before it can be published in Molecules. The specific comments divided in sections are listed below:

General issues:

  • General poor image quality, especially in Figure 4 and panels 1F and 1H.
  • Axis labels are difficult or impossible to read.
  • English correction throughout the manuscript is needed.
  • Please correct typos i.e. lines 148, 149, 251 (Figure 5G) etc.

Section 2.1

  • Cite Figure 1A on the text.
  • Cell lines used for TCN inhibition assay (Figure 1B) should be listed in materials and methods. Which was the criterion for choosing that cell lines?
  • Which is the origin of human normal live cells? Although the IC50 is higher than in any cell line, it is relatively close to the IC50 of some cell lines, like A549. This is really important to validate the possible side effects in the use of this drug.
  • SRB is an abbreviation and therefore should be defined the first time it appears in the text.
  • A reference and/or manufacturer must be provided for the SRB assay.
  • Which was the increase in percentages of the cells in G0/G1? Did you perform more than one experiment to confirm the results?

Section 2.2

  • To support the conclusions drawn from Figure 2, quantification of the bands using an image software must be included.
  • Does the signal in P-EGFR western blot is specific?

Section 2.3

  • To demonstrate the decrease in the binding of STAT3 to target sequences an EMSA assay was used. With this assay you can estimate the quantity of STAT3 able to bind the specific DNA probe in the nuclear extracts, but no the amount of STAT3 bound to chromatin. I think that the correct assay in this case should be a ChIP assay coupled with quantification of the enrichment of several target genes compared with the control.
  • Contradictions can be observed between results in figure 1 and figure 3. In panel 1B you showed that IC50 for HeLa is 2.5 µM of TCN, however no reduction in the viability is reported in panel 3E, where the concentration of TCN reached 4 µM of TCN. Please explain the results. Is the difference associated with the time of treatment?
  • Include the specific residue Y705 in line 152

Round 2

Reviewer 1 Report

The authors have addressed all of the necessary points, with the exception of a few minor points indicated below. The manuscript should be acceptable for publication pending the minor revisions.

Point 18: This is a good point by the authors that the extra “1” in front of the residue numbers is an artifact of the PDB and examining the second crystallographic dimer of STAT3 in 3CWG. However, in the text and figures this can lead to ambiguous interpretations and I would strongly suggest using ASN538 and TRP501 (in the text and Figures) but clearly indicating these labels refer to the second dimer.

There are some instances where the authors use the phrase “hydrophobic bonds” although the correct terminology should be “hydrophobic interactions”.

Figure 4: This figure is still presented in very low quality and should be improved. The image of the STAT3 dimers are stretched/skewed. Also, for Figure 4C, the error bars are only shown on the top half of each data point. The data points should not be connected since the data is not presented as a continuous variable.

Author Response

Point 18: This is a good point by the authors that the extra “1” in front of the residue numbers is an artifact of the PDB and examining the second crystallographic dimer of STAT3 in 3CWG. However, in the text and figures this can lead to ambiguous interpretations and I would strongly suggest using ASN538 and TRP501 (in the text and Figures) but clearly indicating these labels refer to the second dimer.

Response 1:As requested, we have added the actual residue numbers without “1” in brackets with different color beside artifact numbers of the PDB in the figure 4A(line 232-233), and indicated these labels refer to the second dimer in the text and legend of the figure(line 251-252).

There are some instances where the authors use the phrase “hydrophobic bonds” although the correct terminology should be “hydrophobic interactions”.

Response 2:Thank you for your kind reminder. We have corrected “hydrophobic bonds” to “hydrophobic interactions”.

Figure 4: This figure is still presented in very low quality and should be improved. The image of the STAT3 dimers are stretched/skewed. Also, for Figure 4C, the error bars are only shown on the top half of each data point. The data points should not be connected since the data is not presented as a continuous variable.

Response 3:Thank you for your kind reminder. We have further improved the quality of Figure 4, and presented Figure 4C with histogram in the new version.

Reviewer 3 Report

In the new version of the manuscript Qi et al. have addressed all the concerns raised by the initial version and the result is significantly improved. In my opinion, the manuscript in its current state is suitable for its publication in Molecules.

Author Response

Point 1: In the new version of the manuscript Qi et al. have addressed all the concerns raised by the initial version and the result is significantly improved. In my opinion, the manuscript in its current state is suitable for its publication in Molecules.

Response 1:Thank you for your positive comments.